# Novel *Chaphamaparvovirus* in Insectivorous *Molossus molossus* Bats, from the Brazilian Amazon Region

**DOI:** 10.3390/v15030606

**Published:** 2023-02-22

**Authors:** Endrya do Socorro Foro Ramos, Wandercleyson Uchôa Abreu, Luis Reginaldo Ribeiro Rodrigues, Luis Fernando Marinho, Vanessa dos Santos Morais, Fabiola Villanova, Ramendra Pati Pandey, Emerson Luiz Lima Araújo, Xutao Deng, Eric Delwart, Antonio Charlys da Costa, Elcio Leal

**Affiliations:** 1Laboratório de Diversidade Viral, Instituto de Ciências Biológicas, Universidade Federal do Pará, Belem 66075-000, Pará, Brazil; 2Programa de Pos-Graduação REDE Bionorte, Polo Pará, Universidade Federal do Oeste do Pará, Santarém 68040-255, Pará, Brazil; 3Laboratory of Genetics & Biodiversity, Institute of Educational Sciences, Universidade Federal do Oeste do Pará, Santarém 68040-255, Pará, Brazil; 4Department of Agricultural Sciences, School of Veterinary Medicine, University of Amazonia, Santarém 68040-255, Pará, Brazil; 5Laboratory of Virology (LIM 52), Instituto de Medicina Tropical, Universidade de São Paulo, São Paulo 05403-000, São Paulo, Brazil; 6Centre for Drug Design Discovery and Development (C4D), SRM University, Delhi-NCR, Rajiv Gandhi Education City, Sonepat 131029, Haryana, India; 7General Coordination of Public Health, Laboratories of the Strategic Articulation, Department of the Health, Surveillance Secretariat, Ministry of Health (CGLAB/DAEVS/SVS-MS), Brasília 70719-040, Distrito Federal, Brazil; 8Department Laboratory Medicine, University of California San Francisco, San Francisco, CA 94143, USA; 9Vitalant Research Institute, San Francisco, CA 94143, USA

**Keywords:** *Chaphamaparvovirus*, metagenomics, *Molossus*, virome, diversity, bats, Amazon

## Abstract

*Chaphamaparvovirus* (CHPV) is a recently characterized genus of the *Parvoviridae* family whose members can infect different hosts, including bats, which constitute the second most diverse order of mammals and are described worldwide as important transmitters of zoonotic diseases. In this study, we identified a new CHPV in bat samples from the municipality of Santarém (Pará state, North Brazil). A total of 18 *Molossus molossus* bats were analyzed using viral metagenomics. In five animals, we identified CHPVs. These CHPV sequences presented the genome with a size ranging from 3797 to 4284 bp. Phylogenetic analysis-based nucleotide and amino acid sequences of the VP1 and NS1 regions showed that all CHPV sequences are monophyletic. They are also closely related to CHPV sequences previously identified in bats in southern and southeast Brazil. According to the International Committee on Taxonomy of Viruses (ICTV) classification criteria for this species (the CHPV NS1 gene region must have 85% identity to be classified in the same species), our sequences are likely a new specie within the genus *Chaphamaparvovirus*, since they have less than 80% identity with other CHPV described earlier in bats. We also make some phylogenetic considerations about the interaction between CHPV and their host. We suggest a high level of specificity of CPHV and its hosts. Thus, the findings contribute to improving information about the viral diversity of parvoviruses and show the importance of better investigating bats, considering that they harbor a variety of viruses that may favor zoonotic events.

## 1. Introduction

Bats are the second largest order of mammals after rodents; they comprise nearly 20% of all classified mammal species worldwide, with over 1400 species. The order was divided recently into two suborders: *Yinpterochiroptera* (or *Pteropodiformes*), which has five families (i.e., *Rhinopomatidae*, *Rhinolophidae*, *Hipposideridae*, *Craseonycteridae* and *Megadermatidae*) and *Yangochiroptera* (or *Espertilioniformes)* with fourteen families (i.e., *Emballonuridae*, *Furipteridae*, *Miniopteridae*, *Molossidae*, *Mormoopidae*, *Mystacinidae*, *Myzopodidae*, *Natalidae*, *Noctilionidae*, *Nycteridae*, *Phyllostomidae*, *Thyropteridae*, *Cistugidae*, *Vespertilionidae*) [1]. Many bats are insectivores, and most of the rest are frugivores (fruit eaters) or nectarivores (nectar eaters). A few species feed on animals other than insects; for example, vampire bats feed on blood. Most bats are nocturnal, and many roost in caves or other refuges [2]. Since they can fly, it probably enabled them to spread widely into all regions [3]. Apart from the Arctic, the Antarctic and a few isolated oceanic islands, bats exist in almost every habitat on Earth. Tropical areas tend to have more species than temperate ones. They are important in their ecosystems for pollinating flowers and dispersing seeds; many tropical plants depend entirely on bats for these services [2].

Bats are widely regarded as special virus reservoirs due to their association with several important zoonoses, including severe acute respiratory syndrome-related coronavirus (*Coronaviridae*), Nipah henipavirus (*Paramyxoviridae*) and Ebola viruses (Filoviridae) [4,5,6]. However, transmission to humans usually involves an intermediate animal host (such as pigs, camels and horses). Besides, although there is constant contact between humans and bats in some world regions, most viral families that appear regularly in bat virome analysis have not yet spilled over into human populations [7,8].

Potentially, viruses that infect bats can be transmitted to other species by many routes, such as oral-fecal, droplets, airborne and contact with body fluids or tissues [9]. Direct transmission from bats to human has been ascertained by epidemiological and molecular evidence. This was the case of the outbreaks of Nipah virus in Bangladesh associated with the consumption of raw palm tree sap [10]. Other examples of direct bat-to-human spillovers include several Marburg virus outbreaks across Africa [11] and outbreaks of Lyssaviruses globally [12]. Likewise, retrospective serological studies revealed Henipavirus spillover among bat hunters in Cameroon [13] and Filovirus exposure of bat hunters in India [14]. There is indirect bat-to-human spillover involving intermediate hosts, including the Hendra virus in 1994 involving horses [15] and the Nipah virus in Malaysia in 1997 and 1998 related to the infection of pigs [16]. More recently, the SARS-CoV 2002–2003 outbreak in southern China and the 2019 SARS-CoV-2 emergence in central China were retrospectively connected to bat populations and appeared to involve intermediate hosts [17,18]. There is also the possibility that some viruses (arboviruses) are vectored by hematophagous insects from bats to humans and other species [19]. Despite the overlap of ecological niches of bats with many hematophagous arthropods such as Aedes mosquitoes, the potential role for bats as reservoirs for arboviruses is still unclear [19,20]. For example, Dengue virus nucleic acid and antibodies have been detected in wild-caught Mexican and Caribbean bats [21], but experimental infections of *Artibeus* sp. bats with DENV-2, DENV serotypes 1 and 4 resulted in low levels of viremia, low rates of seroconversion and a lack of detection of viral RNA [22,23]. In addition, an extensive survey in 205 bats captured in households from Costa Rica found low frequency and low levels of DEN [20], thus suggesting that bats have little impact as reservoir hosts for DENV.

The distinctiveness of these flying mammals contributes to the zoonotic risk posed by bat viruses. Bats have a wide range of species; they live all over the globe, and some migrate. They are social animals with very large roosting numbers and species co-habitation, which creates an ideal environment for virus transmission [8,24]. It is important to understand how the flight-adapted physiology of bats leads to a significant viral burden [7,25], because the normal pathology observed in humans and experimental animal models in response to viral infection does not appear to occur in bats [26,27]. Bats are sometimes considered silent virus carriers. This could be due to their distinct anti-inflammatory and proinflammatory responses, as well as distinct immunological attributes such as the low number of interferon genes and the fact that interferon genes are always produced in the absence of an activated immune response [28].

In fact, about 75% of human infectious diseases are zoonoses [25,26,27,28,29], of which a considerable proportion are transmitted by either rodents or bats [30,31]. Although bats have more zoonotic viruses per species, the total number of zoonotic viruses found in bats is smaller than in rodents due to the fact that there are almost twice as many rodent species as bat species. As a result, rodents should remain a critical concern as new viral reservoirs [32,33]. However, bats exhibit an unusual immunological response to viral infections [28]. It has been proposed that rapidly transmitting viruses that have evolved with bat immune systems will likely cause increased pathogenicity after emergence in secondary hosts [34]. This may potentially enhance the emergence of occasional epidemics such as SarsCov2 [25,35,36]. Therefore, to further assess the zoonotic potential of virus-infecting bats, a greater understanding of their antiviral responses is required.

In this context, as Brazil is a continental country, recently experiencing vast deforestation in the Amazon region, there is also a great diversity of bats in the country. Brazil bears one of the greatest diversities of bats in the world; there are 184 species, distributed in 72 genera and 9 families [37,38]. According to the Database of bat-associated viruses (http://www.mgc.ac.cn/cgi-bin/DbatVir, accessed on 9 February 2023), there are 755 sequences of viruses identified in bats in Brazil. Majority of these viruses belong to the *Rhabdoviridae* family (60.5%), followed by the *Coronaviridae* family (21.2%) (see Appendix A for details). There are only three sequences of viruses in the *Parvoviridae* family identified so far in Brazilian bats [39,40]. It is known that parvoviruses constitute a highly diversified group of viruses and can infect vertebrate and invertebrate hosts [41]. Therefore, the investigation of this family of viruses in different species need to be better explored [42,43].

In this study we used a metagenomic next generation (NGS) approach to analyze the diversity of viruses in the endemic insectivorous *Molossus molossus* bats captured in the municipality of Santarém, North Brazil. We found a new species of *Chaphamaparvovirus* (CHPV) *Molossus molossus* bats and performed a phylogenetic study to show that there are at least three CHPV species in Brazilian bats. We also raised some hypotheses about the interaction between CHPV and its hosts. These data contribute to improving information on viral diversity in bats and the surveillance of zoonotic diseases in North Brazil.

## 2. Materials and Methods

### 2.1. Sample Collection

Bats were captured in the municipality of Santarém, located in the state of Para in the lower Amazon region. In total, 18 *Molossus molossus* were captured. The velvety free-tailed bat, or Pallas’s mastiff bat (*Molossus molossus*), is a bat species in the family *Molossidae* [44]. They have short, velvety, very soft fur, ranging from dark brown to reddish brown; the whole body or parts of it have white spots in different parts (ears, wing membrane, fur and tail), while in the second there is a total absence of melanin. The tail is thick and free of its membrane, and for this last characteristic they are popularly known as free-tailed bats. Their weight ranges from 10 to 30 g, making them medium-sized among bats. The members of this family in turn are considered by statistics the fastest bats in flight; the females are smaller than the males; they are specialized insectivores [45]. They are found throughout the Americas, from the southern United States (Florida) to Uruguay. They are found from humid forests to urban areas, living under trees, inside caves, house linings, abandoned structures and buildings under construction; this adaptation to urban areas is due to the high presence of insects [46].

We detected in five animals CHPV sequences (i.e., CHPV_O08, CHPV_B08, CHPV_R04, CHPV_F01 and CHPV_R02). Samples were taken from liver (CHPV_F01), spleen (CHPV_B08), kidney (CHPV_R02 and CHPV_R04) and the oral cavity (CHPV_O08). The locations of these samples are indicated in Appendix A.

Animals were euthanized by intramuscular injection of xylazine hydrochloride (1 mg/kg) associated with ketamine hydrochloride (1–2 mg/kg), followed by intracardiac injection of phenobarbital (40 mg/kg).

To carry out this study, we obtained approval from the Ethics Committee on Animal Use of the Universidade Federal do Oeste do Pará (CEUA/UFOPA) under number 0220220128 and from the Biodiversity Authorization and Information System (SISBIO-18313-1) to capture the chiropterans. The necropsy was performed in the Laboratory of Animal Morphophysiology of the Universidade Federal do Oeste do Pará, following the institutional norms of biosecurity.

### 2.2. Processing of the Samples

We initially performed the aliquot of the samples to have enough material to perform all steps of the sequencing protocol applied to viral metagenomics (Next Generation Sequencing) [47].

Organ fragments were first macerated in tissue disruptor, then diluted in 1 mL of Hanks buffered saline solution (HBSS), added to a 2 mL tube containing lysis matrix C (MP Biomedicals, Santa Ana, CA, USA), followed by “rigid rotation” in a Beckman Coulter Optima LE-80 ultracentrifuge with Heraeus Maximum rotor at 32.000 rpm for 5 min to sediment the viral particles. A total of 500 uL of the supernatants was put through 0.45 µM filters (Merck Millipore, Billerica, MA, USA) to remove particles of eukaryotic and bacterial cells. Approximately 100 μL of PEG-it virus precipitation solution (System Biosciences, Palo Alto, CA, USA) was added to the filtrate and the contents of the tube were gently homogenized, followed by incubation at 4 °C for 24 h. After the incubation period, the mixture was centrifuged at 10,000× *g* for 30 min at 4 °C and the supernatant (~300 μL) was discarded. The pellet, rich in virus-like particles (VLPs), was treated with a combination of nuclease enzymes (TURBO DNase and RNase Cocktail Enzyme Mix-Thermo Fischer Scientific, Waltham, MA, USA; Baseline ZERO DNase and RNase-Epicenter, Madison, WI, USA; Benzonase-Darmstadt, Darmstadt, Germany, and RQ1 DNase-Free DNase and RNase A Solution-Promega, Madison, WI, USA) to digest unprotected nucleic acids. The resulting mixture was subsequently incubated at 37 °C for 2 h.

After incubation, viral nucleic acids were extracted using Maxwell^®^ 16 Viral Total Nucleic Acid Purification Kit (Promega, WI, USA) according to the manufacturer’s protocol. The cDNA synthesis was performed with AMV reverse transcription (Promega, WI, USA). A second strand of cDNA synthesis was performed using DNA Polymerase I Large (Klenow) Fragment (Promega, WI, USA).

Viral DNA was enriched by rolling circle amplification (RCA) using the TempliPhi 500 Amplification Kit (Cytiva, CA, USA). Briefly, 1 μL of nucleic acid extract plus 5 μL of the kit sample buffer was heated to 95 °C for 3 min, cooled on ice, and 5 μL of the kit reaction buffer and 0.2 μL of Enzyme mix (Phi29 DNA polymerase) added. The RCA reaction was incubated at 30 °C for 24 h, followed by heat inactivation at 65 °C for 10 min.

Subsequently, duplicated sample (total nucleic acid and RCA) submitted via a Nextera XT Sample Preparation Kit (Illumina, CA, USA) was used to construct a DNA library, identified using dual barcodes. For size range selection, Pippin Prep (Sage Science, Inc.) was used to select a 600 bp insert (range 500–700 bp). The library was deep-sequenced using the NovaSeq 6000 Sequencer (Illumina, CA, USA) with 250 bp ends. Bioinformatics analysis was performed according to the protocol described by Deng et al. [48]. Contigs that shared a percent nucleotide identity of 95% or less were assembled from the obtained sequence reads by de novo assembly. Based on the bioinformatics pipeline used [48], no reads related to human, plant, fungal, or bacterial sequences were obtained. The final near complete genomes were mapped with Geneious R9. All sequences generated in this study were deposited in the Genbank with the accession numbers OQ420631–OQ420635.

### 2.3. Bioinformatics Analysis

The raw reads obtained from Illumina sequencing were pre-processed, where: (a) the paired-end sequence records were removed from both ends, (b) the low-quality sequences (raw data generated from reads less than 100 bp in length) and the adapter and primer sequences were cut using the VecScreen based on BLAST (Basic Local Alignment Search Tool) in the default parameters (MCGinnis and Thomas, 2004) and finally (c) the readings that contained homopolymer and duplicated readings were identified and removed. Bioinformatics data were analyzed according to the previously described protocol [47]. The contigs resulting were compared using BLASTx and BLASTn to search for similarity with viral proteins and nucleotides, respectively, from the GenBank genetic sequence database (http://www.ncbi.nlm.nih.gov, accessed on 21 December 2022). The best results from searches in BLAST (with the highest percentage of sequence similarity already deposited in Genbank) were selected and, to reduce the number of random matches, E values (e-value) were defined in each search.

### 2.4. Genome Annotation

The nucleotide sequences of the genomes and the amino acid sequences of the NS1 and VP1 proteins from the genomes of CHPV detected in this study and other related viruses were selected based on the best results (best hits) of the BLAST search. Using MAFFT tools, complete or nearly complete genomes were aligned [49]. The SF3 Helicase domain and protein motifs of conserved proteins were predicted using InterProScan (https://www.ebi.ac.uk/interpro/search/sequence/, accessed on 10 January 2023) and Motif Finder (https://www.genome.jp/tools/motif/, accessed on 12 January 2023), respectively.

### 2.5. Genetic Distance

The genetic distance and its standard error were calculated using the composite maximum likelihood model plus gamma correction and bootstrap with 1000 replications. Distances were calculated using MEGA software (Version X) [50]. To estimate sequence similarity, a pair-wise method implemented in the SDT program, version 1.2 [51], was used. Estimating the similarity alignments of each unique pair of sequences was performed using algorithms implemented in MUSCLE [52]. After calculating the identity score for each pair of sequences, the program uses the NEIGHBOR component of PHYLIP to calculate a tree. The rooted neighbor joint phylogenetic tree orders all sequences according to their probable degrees of evolutionary relatedness. The results are presented in a frequency distribution of paired identities in a graphical interface.

### 2.6. Phylogenetic Analysis

Phylogenetic trees were constructed using the maximum likelihood approach, and branching support was estimated using a bootstrap test with 1000 replications using the IQ-Tree tool [53]. Trees were visualized and edited using Figtree version 1.4.2 (http://tree.bio.ed.ac.uk/software/figtree, accessed on 9 February 2023).

### 2.7. Likelihood Mapping

Likelihood mapping is an approach that explores the phylogenetic content of a set of aligned sequences [54]. The method is based on an analysis of the maximum likelihoods for the three fully resolved tree topologies that can be computed for four random sequences. The three likelihoods are represented as one point inside an equilateral triangle. The central region of the triangle represents star-like trees (unresolved), vertices represent well-resolved phylogenies and regions connecting the vertices reflect the situation where it is difficult to distinguish between two of the three trees. The location of the likelihoods in the triangle indicates the phylogenetic pattern in the alignment. Thus, likelihood mapping can be used to test a posteriori the phylogenetic signal and the confidence of an inner branch in a phylogenetic tree. The likelihood mapping was performed using the software tree-puzzle version 5.3 (Likelihood-mapping).

### 2.8. Phylogenetic Networks

We used phylogenetic networks to explore reticulated evolution in the CHPV sequences. Networks are graphs used to visualize evolutionary relationships when reticulation events such as hybridization, horizontal gene transfer, recombination, or gene duplication and loss are involved. They differ from traditional phylogenetic trees by the explicit modeling of richly linked networks, by means of the addition of hybrid nodes (nodes with two parents) instead of only tree nodes (a hierarchy of nodes, each with only one parent). They are more realistic models of evolution. Moreover, network methods also provide a value tool for phylogenetic inference even when reticulation events do not play an important role. The combined effect of sampling error and systematic error makes the phylogenetic inference uncertain, and network methods provide tools for representing and quantifying this uncertainty. We used the neighbornet approach implemented in the Splitstree version 4.18.3 [55].

## 3. Results

### 3.1. Characterization of Samples

We were able to retrieve five contigs, near full-length genomes of CHPV, and blast searches indicated that the best cognate reference was NC_032097. This sequences is a CHPV previously identified in hematophagous *Desmodus rotundus* bats in southern Brazil. The genome size of our sequences ranged from 3797 to 4284 bp, consistent with other parvoviruses (<4 kb), with a GC content between 44.03 and 44.89%. BLASTn comparative analysis showed that the sequences described here had a low identity in the NCBI nucleotide database (>80.19%), with coverage of up to 91% with the NC_032097. Descriptions of size and similarity, based on BlastN searches, are summarized in Table 1.

### 3.2. Comparative Analysis of Genomes

In all CHPV sequences described here, the NS1 and VP1 regions were detected, as well as the SF3 Helicase protein domain, characteristic of the *Parvoviridae* family.

To analyze the differences between the sequences, we selected the region of the NS1 protein as it is the most used for taxonomic classification. We compared the amino acid region NS1 of the reference sequences (MG693107, MT734803, JX885610 and NC_032097.1) with those detected in the present study (CHPV_O08, CHPV_B08, CHPV_R04, CHPV_F01, CHPV_R02). The CHPV sequences were detected with three motifs: (I) parvovirus NS1 non-structural protein; (II) papillomavirus helicase; and (III) threonyl carbamoyl adenosine biosynthesis protein TsaE, besides dynein and ATPase domains (AAA7, AAA15 and AAA23). Interestingly, the CHPV sequences described here have an additional motif (DUF3648) with an unknown function (Figure 1). The DUF3648 motif is located between residues 107 and 153 of the NS1 protein of all Brazilian Molossus CHPV (i.e., 107RSDSFKRTLERVWKDVSIVAMSDIELPDPTLEVVKCQKCHKPSSLIS153). We also found an additional ORF within the NS1 region, which seems to be a common feature of all chaphamaviruses [56]. The alignment of Brazilian CHPV identified in *Molossus molossus* is summarized in Appendix A.

### 3.3. Phylogenetic Inference

Initially, we inferred a phylogenetic tree with reference sequences of parvoviruses from the subfamilies of *Parvovirinae*, *Densovirinae* and *Hamaparvovirinae*, using complete genomes (tree not shown). The tree shows that all the CHPV sequences of this study are monophyletic and clustered in the subfamily *Hamaparvovirinae* genus *Chaphamaparvovirus.* This monophyletic group is in a clade formed by sequences identified in bats (NC_032097 identified in *Desmodus rotundus* and MH170363_ identified in *Artibeus lituratus*). Next, to obtain more resolution we used only complete genomes of chaphamapavoviruses and constructed a maximum likelihood tree (Figure 2). This tree shows multiple clusters with high bootstrap support. Most of these clades composed by a unique kind of host were collapsed to facilitate the visualization (i.e., Feline, Canine and Avian). The topology of the tree also indicates that all CHPV from Molossus bats are monophyletic (delineated by a blue rectangle). Another important feature of the tree is the clade formed by CHPV identified in Brazilian bats with high support.

To better clarify the relatedness of our sequences and other CHPV references, we evaluated the phylogenetic signal of the alignments and constructed trees using VP1 (Capsid) and the NS1 genomic regions. Initially, we show that the phylogenetic signal in alignment with VP1 and NS1 regions is better than the signal in the alignment of the complete genome. To obtain these results we used the likelihood mapping approach [54]. This analysis is based on the assumption that by plotting the likelihood of all possible quartets on a three-dimensional coordinate system (triangular surface) it is possible to determine the number of unresolved quartets (star-like trees). Ideally, fully resolved quartets will occupy vertices of the triangular surface whilst star-like trees will occupy the center of the triangle. In this regard, alignments containing a strong phylogenetic signal will provide likelihood maps in which the proportion of plotted likelihoods in the center of the triangle is inferior to the sum of the likelihoods plotted in the three vertices (see the diagram in Figure 3a). Our results showed that the NS1 regions are the best genomic regions for performing a phylogenetic study since they have 43.1% fully resolved quartets compared with the 42.2% of star trees (Figure 3b). This is probably because VP1 are highly variable regions with many invariable sites and some deletions/insertions.

Since the NS1 regions of CHPV have the better phylogenetic signal, we used these regions and constructed a maximum likelihood tree to obtain more details about the relatedness of these viruses. Likewise, the NS1 tree shows many well supported clades are composed by CHPV identified in a same category of host (Figure 4). For example, the clade Feline is composed of sequences identified in domestic cats from distinct regions (Figure 4). In the same way, the cluster Brazilian bats is equally composed of CHPV identified in bats from Brazil. Moreover, CHPV identified in certain categories of birds tend to cluster together, such as psittacines and passerines.

### 3.4. Reticulated Evolution of CHPV

Previously, we found a high percentage of unresolved trees in CHPV sequences. This suggests that the evolutionary history of these strains is complex. Then, we decided to investigate the evolutionary scenario of CHPV sequences using a network approach. Phylogenetic analysis provides a scenario where two related sequences share a hypothetical ancestor, thus generating a bifurcating tree. The network method, on the other hand, provides a general description of the complexity of the evolutionary history of sequences; it generates reticulated graphs in which the edges represent the evolutionary connection among sequences. The results indicated by the network reveal some level of reticulated evolutionary events among the main CHPV clades (Figure 5). We found some reticulated edges (indicated by arrows) connecting the groups Bats and Rodent. There are also many reticulations among sequences in the group avian. This is in concert with the phylogenetic analysis that showed the avian clade is composed by subclusters (passeriformes, psittaciformes piciformes, galliformes, anseriformes and gruiformes.

### 3.5. Genetic Diversity

We used the nucleotide similarity of NS1 to compare the variability of our sequences and some CHPV references. For this comparison we used CHPV sequences from bats plus sequences from the rodent clade: one sequence identified in capuchin kidney; one identified in Ursus americanus; one identified in Tasmanian devil (*Sarcophilus harrisii*) and one CHPV sequence identified in Gulf pipefish (*Syngnathus scovelli*). The pairwise identity matrix indicates that the mean identity among CHPV identified in Molossus bats is 96% (gray rectangle in Figure 6a). In general, the pairwise identities of CHPV from different hosts ranged from 58% to 79%. The percentage of NS1 identity between CHPV identified in *Molossus molossus* and CHPV identified in *Desmodus rotundus* is 76% while the identity with CHPV identified in *Artibeus lituratus* is 78% (Figure 6b). It is important to mention that the overall genomic nucleotide diversity among the *Molossus molussus* CHPV is 0.1%. In contrast, the general diversity of CHPV identified in bats is higher than 50%.

## 4. Discussion

Previous investigations in Brazil have already reported the identification of many viruses in bats. Most studies, however, have been performed in southern Brazil [40,42]. Viral investigations of *Molossus molossus*, *Artibeus lituratus* and *Sturnira lilium* indicated that these bats are commonly infected by Flavivirus, Coronavirus, Arenavirus, Paramyxovirus, Adenovirus, Papillomavirus and Parvovirus [42,46]. This shows the extreme importance of viral surveillance in bats [29].

Here, we report the molecular characterization of CHPV sequences identified in bats captured in the northern region of Brazil. We found five closely related CHPV sequences retrieved through the NGS of samples (oral swab, spleen, liver and kidney) from bats of the *Molossus molossus* species.

Parvoviruses have a genome consisting of linear single-stranded DNA (ssDNA) with terminal repeat structures [41]. The genome encodes two open reading frames (ORFS) that encode the NS1 proteins associated with the replication region (rep) and the Vp1 protein related to the capsid region (cap) [58]. They belong to the *Parvoviridae* family, which is subdivided into three subfamilies *Parvovirinae*, *Densovirinae* and *Hamaparvovirinae* [59,60]. The subfamily *Hamaparvovirinae* is divided into five genres: *Brevihamaparvovirus*, *Chaphamaparvovirus* (CHPV), *Hepanhamaparvovirus*, *Ichthamaparvovirus* and *Penstillaparvovirus* [59].

The genus *Chaphamaparvovirus* contains 16 species, all described through metagenomic investigations. Members have been described since 2012 [59] in various hosts such as rodents [60], birds [61,62,63], reptiles [64], turkeys [65], pigs [66], fish [67], dogs and cats [68,69,70]. Furthermore, CHPVs have already been detected in human samples [71]. *Chaphamaparvovirus* has also been identified in different geographic regions (Australia, Brazil, Canada, China, France, Holland and Switzerland), hosts and samples, such as in human plasma samples [59,60,61,62,63,64,65,66,67]. Nevertheless, molecular, epidemiological and pathogenic characteristics have not yet been fully established [65].

We performed a detailed phylogenetic analysis using the complete genome and NS1 region. The results showed that the sequences identified in this study are monophyletic and belong to the same CHPV lineage because they have 96% of identity in the NS1 gene region. Phylogenetic trees also show that the closest CHPV to our sequences are closely related with sequences previously identified in hematophagous *Desmodus rotundus* bats in southeast Brazil [40] and insectivorous *Artibeus lituratus* bats from South Brazil [72]. The median identity between our sequences and the *Desmodus rotundus* CHPV is 76%. The *Desmodus rotundus* is a hematophagous bat that is endemic in Brazil [42,48]. *Desmodus rotundus* bats have a broad distribution and can live in urban and in rural areas as well [73]. Although these bats have distinct feeding behaviors, they cohabit in many regions, and colonies of these bats can share resting places such as caves and abandoned buildings.

We found that the median identity between our sequences and the *Artibeus lituratus* CHPV was 78%. The bats are frugivorous and also endemic in Brazil [74]. It is important to mention that the NS1 nucleotide identity between *Desmodus rotundus* CHPV and *Artibeus lituratus* CHPV is 81%. Accordingly, the amino acid identities of our sequences with *Desmodus rotundus* CHPV and *Artibeus lituratus* CHPV are 76% and 69%, respectively. Likewise, the identity between *Desmodus rotundus* CHPV and *Artibeus lituratus* CHPV is 71%. The criteria adopted by the ICTV for parvovirus classification at the species level specify that amino acid identity of the NS1 protein lower than 85% is enough to define a new species [41]. Another feature of the CHPV in *Molossus molossus* bats is the presence of the DUF3648 motif, which is a unique signature of the VP1 of these bats. Consequently, the CHPV identified in *Molossus molossus* bats are indeed a new species.

Another observation of our analysis of CHPV sequences is the clustering pattern observed in trees inferred with complete genomes or the genes VP1 and NS1. In all trees, there are some phyloclades with high support, and these clades are mostly composed by CHPV identified in the same category of animals. For example, the Canine clade has 14 sequences from members of the *Canidae* family (dog, wolf and coyote) and two sequences identified in domestic cats. The same pattern is observed in the passeriforme and psittaciforme clades. In the same way, CHPV identified in Brazilian bats cluster together. Although it is not possible to determine the natural hosts of viral sequences generated by metagenomic studies, the phylogenetic evidence of host–virus association can be used as a proxy in this association. Besides, the clustering networks also indicate that CHPV isolated in certain species are closely related. Therefore, it is tempting to suggest that there is a correlation between CHPV and its hosts, with eventual cross-species transmissions. If the clustering pattern observed in CHPV is due to virus–host specificity, then it is in contrast with the plasticity of carnivorous protoparvovirus to infect multiple species. Protoparvovirus naturally infect a plethora of hosts in the order Carnivora. They share 98% identity in the nucleotide sequence and have common ancestors in the recent past. For example, the feline panleukopenia virus (FPV) infects domestic cats (*Felis catus*) and other carnivores including the American mink (*Neovison vison*), raccoon (*Procyon lotor)* Puma (*Puma concolor*). The FPV sequences of lineages isolated from distinct hosts intermingle in phylogenetic trees, indicating that this virus easily spreads among distinct species. In the same way, the canine parvovirus (CPV) infects domestic dogs (*Canis lupus familiaris)*, wolves (*Canis lupus)* and coyotes (*Canis latrans*) [75,76]. Conversely, our phylogenetic analysis showed that evolutionary clades in CHPV are mostly structured according to the host in which this virus was identified. Although CHPV share a remote ancestry with other parvoviruses and some features of their genomes are conserved, the host adaptive evolution of CHPV might differ from other members of the *Parvoviridae* family. *Chaphamaparvovirus* was recently identified by metagenomic virus studies and there are low numbers of sequences originated from independent studies in public databases. Little is known about the life cycle, host range and cell tropism of this virus. To better understand the evolutionary complexity of CHPV, further studies need to be performed in order to increase the number of CHPVs identified in different species.

## 5. Conclusions

The introduction of sequencing platforms associated with a metagenomic approach to the identification of new viruses, including potentially zoonotic ones, has become more frequent. For example, SARS-CoV-2 probably originated from a zoonotic event involving bats and other unknown species. Since bats are natural reservoirs of many pathogens and Brazil has one of the greatest bat diversities, viral surveillance, especially of bats living near humans, is extremely important to expand our epidemiological and demographic knowledge regarding viruses infecting bats.

## Figures and Tables

**Figure 1 viruses-15-00606-f001:**
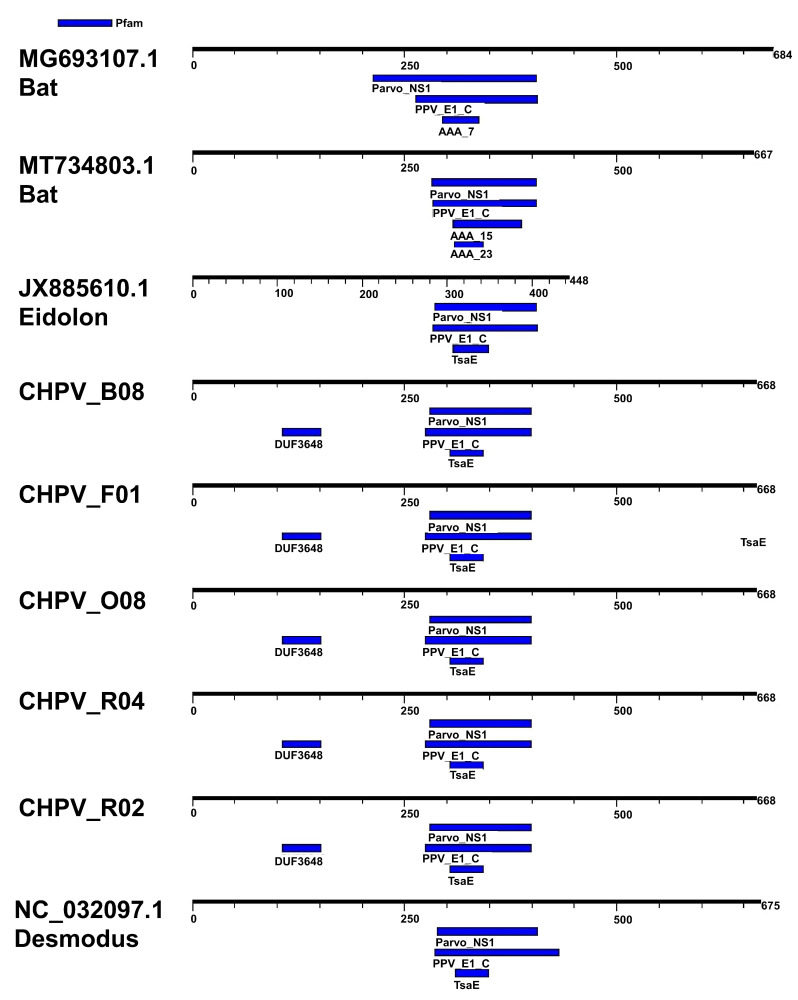
Motifs in the CHPV non-structural protein (NS1). Figure shows the motifs detected in the NS1 protein of CHPV. Motifs were detected using the server MotifFinder and the Pfam database was used as reference (https://www.genome.jp/tools/motif/, accessed on 22 January 2023).

**Figure 2 viruses-15-00606-f002:**
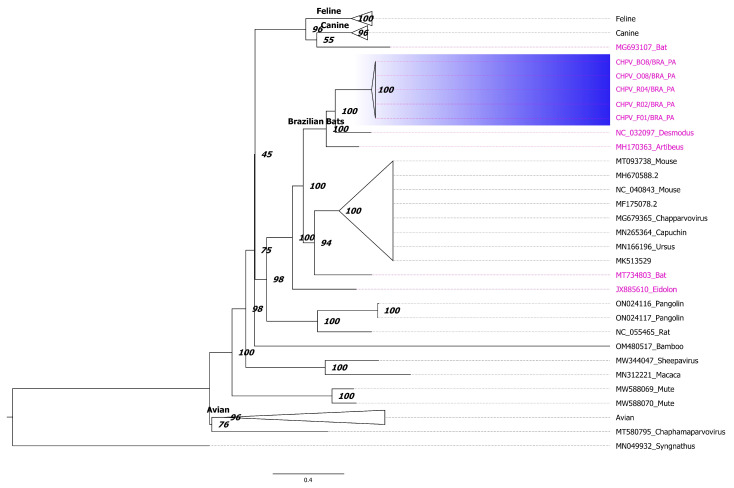
Maximum likelihood tree of chaphamaparvoviruses. The tree was constructed using complete CHPV. Some clades composed by sequences isolated from the same host (i.e., Feline, Canine and Avian) were collapsed to facilitate the visualization. Sequences identified in bats are indicated by magenta color. The tree is unrooted and numbers in the nodes represent bootstrap values. Horizontal line under the tree indicates the number of nucleotide substitutions per site. The tree was constructed using GTR model plus gamma correction and the proportion of invariable sites was estimated. Bootstrap values were calculated using the ultra-fast approximation approach [57].

**Figure 3 viruses-15-00606-f003:**
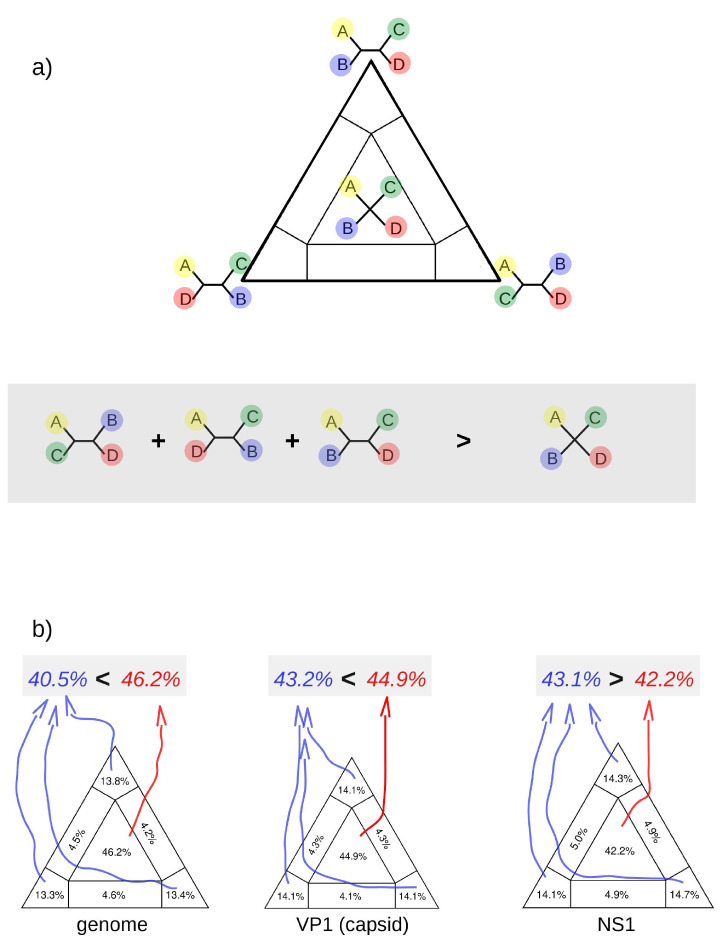
Maximum likelihood map. (**a**) Schematic representation of the maximum likelihood map approach. In the triangular surface, individual likelihoods of all possible quartets will be plotted. The likelihoods of resolved quartets occupy vertices of the triangle and the likelihoods of unresolved trees will occupy the center of the surface. Alignments with proper phylogenetic signals generate likelihood maps in which the proportion of plotted likelihoods in the center of the triangle is lower than the sum of likelihoods in all vertices (diagram in the gray area). (**b**) Likelihood maps of the genome, VP1 and NS1 regions of parvoviruses.

**Figure 4 viruses-15-00606-f004:**
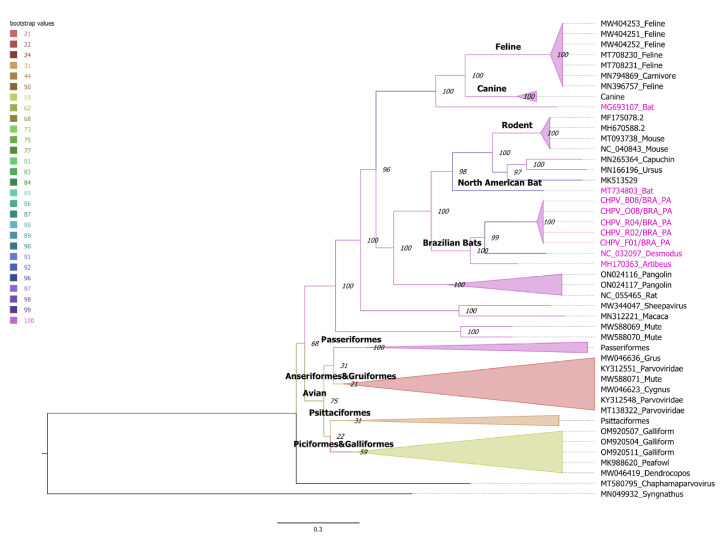
Maximum likelihood NS1 tree. Numbers above branches are bootstrap values. Sequences identified in bats are indicated by magenta color. The tree is unrooted and numbers in the nodes represent bootstrap values. Branches are colored according to the bootstrap support following color scale indicated in the figure. Horizontal line under the tree indicates the number of nucleotide substitutions per site. Clades in which CHPV was detected in the same category of host are indicated by names (Avian, Rodent, Bats, Feline and Canine). The tree was constructed using the GTR model plus gamma correction and the proportion of invariable sites was estimated. Bootstrap values were calculated using the ultra-fast approximation approach [57].

**Figure 5 viruses-15-00606-f005:**
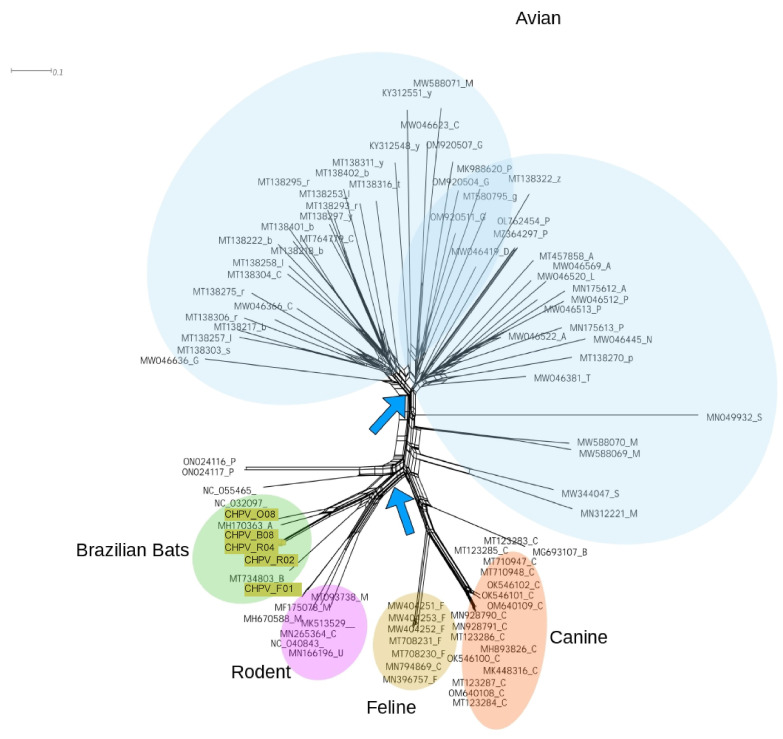
Network graph of the NS1 genomic region of CHPV. The diagram shows a network clustering similar sequences. The main clusters are indicated by different colors. Sequences from this study are highlighted. Arrows indicate reticulated edges connecting sequences. Distances were calculated assuming the HKY85 evolutionary model. The network was constructed using the neighbornet approach implemented in Splitstree version 4.18.3 [55].

**Figure 6 viruses-15-00606-f006:**
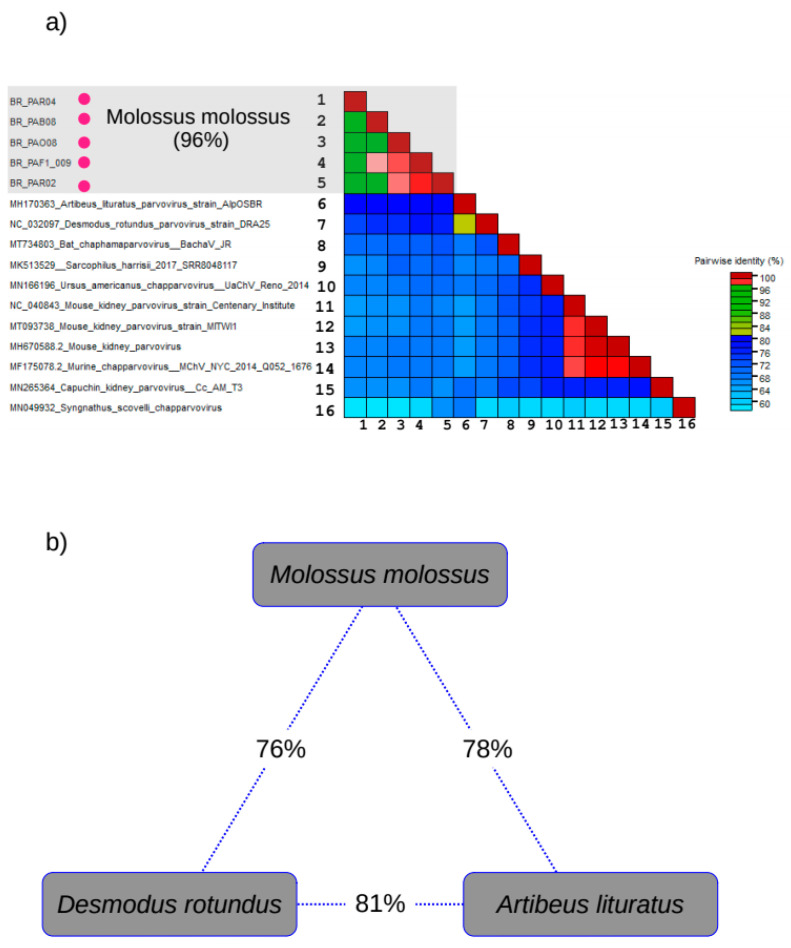
Nucleotide identity of NS1 of CHPV. (**a**) Identity matrix showing pairwise comparisons among distinct CHPV sequences. (**b**) Chart depicting the pairwise identity of NS1 gene in CHPV identified in Brazilian bats.

**Table 1 viruses-15-00606-t001:** Nucleotide identity of CHPV sequences identified in *M. molossus*.

Name	Size (Base Pairs)	Coverage	Nucleotide Identity	E-Value
CHPV_O08	4284	89%	78.71%	0.0
CHPV_B08	3797	89%	80.19%	0.0
CHPV_R04	3800	91%	79.61%	0.0
CHPV_F01	4281	90%	78.91%	0.0
CHPV_R02	4001	90%	79.54%	0.0

ID: Percentage of identity between our CHPV sequences and the CHPV reference NC_032097.

## Data Availability

Not applicable.

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
