# Peer review of "Novel Chaphamaparvovirus in Insectivorous Molossus molossus Bats, from the Brazilian Amazon Region"

_viruses, 2023, doi:10.3390/v15030606_

Round 1
Reviewer 1 Report
The paper describes novel chaphamaparvoviruses identified from bats from Brazil using viral metagenomics. The authors justify the need for the surveillance of bats for potential zoonotic disease outbreaks. Methods are sound and described adequately. Results have been well presented, and diagrams are shown clearly. Appropriate citations are given. The manuscript is clear and well-structured. The manuscript can be accepted in its current form, with one mistake corrected:
Line 273: genera ILO gener
Author Response
We changed the new version of the manuscript in order to make it clear to the readers.
Reviewer 2 Report
This is study that adds to the growing body of data related to bat viruses.
However, the manuscript needs significant improvement. I do not recommend publishing the manuscript in the form in which it is presented now. Main disadvantages:
1. The authors sequenced incomplete genomes of viruses, the only contigs. The authors says that 89-90% of the genome has been determined. With an expected genome length up to 4000-5000, this means that only 500-700 bp remain undetermined. This is quite a bit. Please finishe genomes. For example, the missing regions can be sequenced using the Sanger method (not NGS).
- If it is not possible to complete genomes assembling, then please provide information on which areas remained uncovered. Perhaps this is a similar region of the genomes?
2. Why does the introduction give detailed information about what these types of bats look like? This information is not used later in the article, and indeed it is not clear how it relates to the viruses narrative.
3. Table 1.
E-value equal to 0.0 is werde. It is impossible. Please check your results.
4. Figure 2. Motifs in the CHPV
The picture as it presented is redundant. The structure of all your viruses is the same. Why are you drawing five completely identical schemas? I mean schemas which are highlighted in blue. All these schemes can be reduced to one, which will be common for all your viruses.
5. There is no conclusion
Author Response
This is study that adds to the growing body of data related to bat viruses.
However, the manuscript needs significant improvement. I do not recommend publishing the manuscript in the form in which it is presented now. Main disadvantages:
1. The authors sequenced incomplete genomes of viruses, the only contigs. The authors says that 89-90% of the genome has been determined. With an expected genome length up to 4000-5000, this means that only 500-700 bp remain undetermined. This is quite a bit. Please finishe genomes. For example, the missing regions can be sequenced using the Sanger method (not NGS).
- If it is not possible to complete genomes assembling, then please provide information on which areas remained uncovered. Perhaps this is a similar region of the genomes?
RESP: We appreciate this comment. Regions not fully complete are most located at the extremities of sequences. Since we used DeNovo assembly we are not sure if these sequences are not finished. We included a fire (figure S2) showing the differences in the alignment of CHPV identified in Mololussus Bats.
2. Why does the introduction give detailed information about what these types of bats look like? This information is not used later in the article, and indeed it is not clear how it relates to the viruses narrative.
RESP: The introduction section was modified in this new version of the manuscript.
3. Table 1.
E-value equal to 0.0 is werde. It is impossible. Please check your results.
RESP: E-values (like p-values) give the probability of observing events by chance. So, 0.0 is the possibility of finding by chance a match between our sequences (query) and the best hit indicated by the Blast search.
4. Figure 2. Motifs in the CHPV
The picture as it presented is redundant. The structure of all your viruses is the same. Why are you drawing five completely identical schemas? I mean schemas which are highlighted in blue. All these schemes can be reduced to one, which will be common for all your viruses.
RESP: Indeed they show identical maps but each map represents a different chaphamaparvovirus.
5. There is no conclusion
RESP: In this version, a conclusion section was included.
Reviewer 3 Report
Brazil is a country with the greatest bat diversity and bats serve as an important reservoir of viruses. Therefore, it is essential to investigate bat viruses. In this manuscript, five strains of chaphamaparvovirus were detected in bats in parts of Brazil. Genomic analysis, phylogenetic analysis and genetic distances reveal that these five strains are a new lineage distinct from other bat chaphamaparvovirus. This manuscript provides further information in understanding the diversity of chaphamaparvovirus in bats. However, the manuscript does not provide deeper insights about chaphamaparvovirus, The valid information provided is limited using bioinformatics analysis.
1. Line 62-77, a large amount of text to describe the details informations of the bats is not necessary in a a study focusing on viruses. However, a brief description of the role of bats in virus transmission and the current status of bats species in Brazil is desirable.
2. The introduction of the manuscript has too many subparagraphs and should be re-constructed to give it a reasonable paragraph order.
3. Line101-102,
3. Figure 1 is not necessary to appear in the text, it can be submitted as supplementary material.
4. Did you upload the sequence to Genbank or another database? if so, the sequence ID should be listed in the results or in the “Data Availability Statement”.
5. Line 227, “M. molossus” should be revised to italicize.
6. Line 239-240, I don’t think this point is significant.
7. Figure 4, whether “VP” should be modified to “NS”, please consider.
8. Line 301, “South and Southern” may not correct.
9. Line 328, As far as I know, there are not many sequences of chaphamaparvovirus in bats, so this statement is inappropriate due to the limitations of the sequence information. Please revise it to a more conservative sentence or a forward-looking sentence.
Author Response
REV3
Brazil is a country with the greatest bat diversity and bats serve as an important reservoir of viruses. Therefore, it is essential to investigate bat viruses. In this manuscript, five strains of chaphamaparvovirus were detected in bats in parts of Brazil. Genomic analysis, phylogenetic analysis and genetic distances reveal that these five strains are a new lineage distinct from other bat chaphamaparvovirus. This manuscript provides further information in understanding the diversity of chaphamaparvovirus in bats. However, the manuscript does not provide deeper insights about chaphamaparvovirus, The valid information provided is limited using bioinformatics analysis.
1. Line 62-77, a large amount of text to describe the details informations of the bats is not necessary in a a study focusing on viruses. However, a brief description of the role of bats in virus transmission and the current status of bats species in Brazil is desirable.
RESP: We changed the introduction and added a description about transmission of viruses from bat-to-man.
2. The introduction of the manuscript has too many subparagraphs and should be re-constructed to give it a reasonable paragraph order.
RESP: We changed the introduction in order to make it clearer and concise.
3. Line101-102,
3. Figure 1 is not necessary to appear in the text, it can be submitted as supplementary material.
RESP: This figure is in the supplementary material now.
4. Did you upload the sequence to Genbank or another database? if so, the sequence ID should be listed in the results or in the “Data Availability Statement”.
RESP: We did include the Genbank IDs of all sequences generated in our study. We state there is no data to be available in this study.
5. Line 227, “M. molossus” should be revised to italicize.
RESP: This was reviewed in the text. When we mention Molussus CHPV or Molussus bats there is no need to italicize.
6. Line 239-240, I don’t think this point is significant.
RESP: This was excluded in the new version of the manuscript
7. Figure 4, whether “VP” should be modified to “NS”, please consider.
RESP: We review the figures and the text. VP (or capsid) protein and NS1 (non structural or polymerase) protein. The NS1 is the region with higher phylogenetic signal.
8. Line 301, “South and Southern” may not correct.
RESP: We change this in the text.
9. Line 328, As far as I know, there are not many sequences of chaphamaparvovirus in bats, so this statement is inappropriate due to the limitations of the sequence information. Please revise it to a more conservative sentence or a forward-looking sentence.
RESP: We also change this text to better clarify the manuscript.
Round 2
Reviewer 3 Report
The authors resubmitted a high-quality manuscript with increased content in the article. However, the article is slightly lengthy and there are some minor problems. Please refer the author to a few points for revision, and if you think it is a wrong suggestion, please ignore it.
1. Line 101-128 In this paragraph the authors want to describe the behavioral characteristics and physiological properties of bats that affect the spread of the virus. However, this paragraph is too long and it is suggested to make it more concise.
2. Line 253 In the new manuscript, there is no result of Genetic distance. The method of Genetic Diversity should be complemented .
3. Line 264 It is suggested that this section should be supplemented with detailed methods on Maximum likelihood map and Network graph. Please recheck the consistency of results and methods.
4. Line 323-327 “Sequences identified in bats are indicated by magenta color in the tree.........collapsed to facilitate the visualization “ It would be better to delete these sentences to make the main text of the article more concise.
5. Line 345-353 As these details pertain to the method rather than the results, it is recommended that the authors provide a comprehensive description of the method in the Materials and Methods section, along with an explanation of the rationale for its use.
6. Line 395-399 These details on methods and principles are best described in Materials and Methods.
7. Line 435 “South and Southern” I'm not sure if these words are a clerical error.
8. Line 38, 460-461 Beacese authors deleted the tree of VP1 and SF3, “VP1, and SF3 Helicas” should be deleted, I think.
9. Line 515 While there may be varying opinions on the appropriate use of references in the conclusion section, it is generally recommended to focus on summarizing the key results of the article, as well as providing an overview of its significance, limitations, and any relevant future directions. Sentences requiring references are typically more appropriate for the discussion section, and the authors are encouraged to consider this in their revisions.
Author Response
The authors resubmitted a high-quality manuscript with increased content in the article. However, the article is slightly lengthy and there are some minor problems. Please refer the author to a few points for revision, and if you think it is a wrong suggestion, please ignore it.
1. Line 101-128 In this paragraph the authors want to describe the behavioral characteristics and physiological properties of bats that affect the spread of the virus. However, this paragraph is too long and it is suggested to make it more concise.
Resp: We changed this paragraph trying to make it clearer.
2. Line 253 In the new manuscript, there is no result of Genetic distance. The method of Genetic Diversity should be complemented .
Resp: We included a brief results regarding the genetic distance that were calculated using genomes of CHPV identified in bats.
3. Line 264 It is suggested that this section should be supplemented with detailed methods on Maximum likelihood map and Network graph. Please recheck the consistency of results and methods.
Resp: We included a explanation about phylogenetic networks and likelihood mapping in the M&M section.
4. Line 323-327 “Sequences identified in bats are indicated by magenta color in the tree.........collapsed to facilitate the visualization “ It would be better to delete these sentences to make the main text of the article more concise.
Resp: These sentences were excluded from the text.
5. Line 345-353 As these details pertain to the method rather than the results, it is recommended that the authors provide a comprehensive description of the method in the Materials and Methods section, along with an explanation of the rationale for its use.
Resp: A description of the maximum likelihood mapping was included in the M&M section of the new version of the manscirpt.
6. Line 395-399 These details on methods and principles are best described in Materials and Methods.
Resp: We included in M&M section a brief description of maximum likelihood mapping and phylogenetic networks as well.
7. Line 435 “South and Southern” I'm not sure if these words are a clerical error.
Resp: We corrected it to southern Brazil.
8. Line 38, 460-461 Beacese authors deleted the tree of VP1 and SF3, “VP1, and SF3 Helicas” should be deleted, I think.
Resp: We appreciate this comment. This was removed from the new text.
9. Line 515 While there may be varying opinions on the appropriate use of references in the conclusion section, it is generally recommended to focus on summarizing the key results of the article, as well as providing an overview of its significance, limitations, and any relevant future directions. Sentences requiring references are typically more appropriate for the discussion section, and the authors are encouraged to consider this in their revisions.
Resp: We appreciate this comment. These references were replaced to the introduction section.